# Predicting CTCF-mediated chromatin interactions by integrating genomic and epigenomic features

Yan Kai [1,2,3], Jaclyn Andricovich[2,3], Zhouhao Zeng[1], Jun Zhu [4], Alexandros Tzatsos[2,3] & Weiqun Peng[1]

The CCCTC-binding zinc-finger protein (CTCF)-mediated network of long-range chromatin interactions is important for genome organization and function. Although this network has been considered largely invariant, we find that it exhibits extensive cell-type-specific interactions that contribute to cell identity. Here, we present Lollipop, a machine-learning framework, which predicts CTCF-mediated long-range interactions using genomic and epigenomic features. Using ChIA-PET data as benchmark, we demonstrate that Lollipop accurately predicts CTCF-mediated chromatin interactions both within and across cell types, and outperforms other methods based only on CTCF motif orientation. Predictions are confirmed computationally and experimentally by Chromatin Conformation Capture (3C). Moreover, our approach identifies other determinants of CTCF-mediated chromatin wiring, such as gene expression within the loops. Our study contributes to a better understanding about the underlying principles of CTCF-mediated chromatin interactions and their impact on gene expression.

---

[1] Department of Physics, George Washington University (GWU), Washington, DC 20052, USA. [2] Department of Anatomy and Cell Biology, Cancer Epigenetics Laboratory, GWU, Washington, DC 20052, USA. [3] GWU Cancer Center, GWU School of Medicine and Health Sciences, Washington, DC 20052, USA. [4] Systems Biology Center, National Heart Lung and Blood Institute, National Institute of Health, Bethesda, MD 20892, USA. Correspondence and requests for materials should be addressed to A.T. (email: atzatsos@gwu.edu) or to W.P. (email: wpeng@gwu.edu)

Higher-order chromatin structure plays a critical role in gene expression and cellular homeostasis[1–7]. Genome-wide profiling of long-range interactions in multiple cell types revealed that CCCTC-binding factor (CTCF) is bound at loop anchors and enriched at the boundaries of topologically associating domains (TADs)[8–11], suggesting that it plays a central role in regulating the organization and function of the 3D genome[12,13]. Depletion of CTCF revealed that it is required for chromatin looping between its binding sites and insulation of TADs[14,15], and disruption of individual CTCF-binding sites deregulated the expression of surrounding genes[16–19]. Mechanistically, many of the CTCF-mediated loops define insulated neighborhoods that constrain promoter–enhancer interactions[13], and in some cases CTCF is directly involved in promoter–enhancer interactions[9,10,20].

The CTCF-mediated interaction network has been considered to be largely invariant across cell types. However, in studies of individual loci, cell-type-specific CTCF-mediated interactions were found to be important in gene regulation[17,21]. Furthermore, CTCF-binding sites vary extensively across cell types[22,23]. These findings suggest that the repertoire of CTCF-mediated interactions can be cell-type-specific, and it is necessary to understand the extent and functional role of cell-type-specific CTCF-mediated loops. If cell-type-specific interactions are prevalent and contribute to cellular function, it would be inappropriate to use the CTCF-mediated interactome derived from a different cell type.

CTCF-mediated loops can be mapped through Chromatin Conformation Capture (3C)-based technologies[2]. Among them, Hi-C[9,24] provides the most comprehensive coverage for identifying looping events. However, it requires billions of reads to achieve kilobase resolution[9]. On the other hand, Chromatin Interaction Analysis using Paired-End Tags (ChIA-PET) increases resolution by only targeting chromatin interactions associated with a protein of interest[10,25,26]. Recently developed protocols, including Hi-ChIP[27] and PLAC-seq[28], improved upon ChIA-PET in sensitivity and cost-effectiveness. Despite recent technical advances, experimental profiling of CTCF-mediated interactions remains difficult and costly, and few cell types have been analyzed[9,10,24,29]. Therefore, computational predictions that take advantage of the routinely available ChIP-seq and RNA-seq data is a desirable approach to guide the interrogation of the CTCF-mediated interactome for the cells of interest.

Here, we carry out a comprehensive analysis of CTCF-mediated chromatin interactions using ChIA-PET data sets from multiple cell types. We find that CTCF-mediated loops exhibit widespread plasticity and the cell-type-specific loops are biologically significant. Motivated by this observation, we develop Lollipop—a machine-learning framework based on random forests classifier—to predict the CTCF-mediated interactions using genomic and epigenomic features. Lollipop significantly outperforms methods based solely on convergent motif orientation when evaluated both within individual and across different cell types. Our predictions are also experimentally confirmed by 3C. Moreover, our approach identifies other determinants of CTCF-mediated chromatin wiring, such as gene expression within the loop.

## Results

**CTCF-mediated loops exhibit cell-type specificity.** We used the ChIA-PET2 pipeline[30] and analyzed published ChIA-PET data sets from three cell lines (Supplementary Table 1): GM12878 (lympho-blastoid)[10], HeLa-S3 (cervical adenocarcinoma)[10], and K562 (chronic myelogenous leukemia)[29]. By using false discovery rate (FDR) ≤0.05 and paired-end tag (PET) number ≥2, we identified 51,966, 16,783, 13,076 high-confidence chromatin loops for GM12878, HeLa, and K562, respectively (Supplementary Table 2). A significant fraction of loops was found to be cell-type-specific (67.9%, 26.2%, and 21.5% of loops in GM12878, HeLa, and K562, respectively (Fig. 1a)). Of note, the GM12878 library has higher sequencing depth, which may contribute to the higher number of identified loops and cell-type-specific loops (Supplementary Table 2 and Supplementary Fig. 1a).

To elucidate what contributes to this plasticity, we compared the CTCF-binding sites identified in ChIA-PET data sets across the three cell lines. We found that only 36% of CTCF binding sites are constitutive (i.e., "+++", Fig. 1b), consistent with previous reports[22,23]. Besides cell-type-specific binding sites, rewiring of shared binding sites also contributes and accounts for 24–44% of the cell-type-specific loops (Fig. 1c and Supplementary Fig. 1b).

**Cell-type-specific loops contribute to gene regulation.** Loops shared among different cell types exhibit significantly higher interaction strength than the cell-type-specific loops (Supplementary Fig. 1c), questioning whether the latter are biologically relevant. To address this question, we asked whether these loops are involved in gene regulation.

First, we found that cell-type-specific loops harbor a significantly higher ratio of tandem CTCF motif orientation compared to shared loops (Supplementary Fig. 1d). This suggests their involvement in gene regulation, given that tandem loops exhibit more regulatory potential than convergent ones[10].

Second, we asked whether cell-type-specific Super-Enhancers (SEs)[31,32] are associated with cell-type-specific loops. SEs regulate cell identity, development, and cancer[31–33], and CTCF was shown to play a critical role in their hierarchical organization[34]. Motivated by these findings, we first carried out Disease Ontology analysis on SEs for each cell type using GREAT[35], confirming that they are linked with the corresponding disease origin (Supplementary Fig. 1e). Next, we compared SEs in HeLa and K562 and identified three sets: HeLa-specific, common, and K562-specifc. HeLa-specific SEs are preferentially associated with HeLa-specific loops, compared to common SEs (Fig. 1d, left panel). Similarly, K562-specific SEs are preferentially associated with K562-specific loops compared to common SEs (Fig. 1d, left panel). The same conclusion was reached when we compared GM12878 vs HeLa as well as GM12878 vs K562 (Fig. 1d, central and right panels). Taken together, cell-type-specific SEs are more likely to be associated with loops specific to that cell type, suggesting the functional significance of cell-type-specific loops.

Third, we examined how cell-type-specific loops are associated with gene expression changes. We found that genes associated with cell-type-specific loops have higher expression levels in the respective cell type. In contrast, there is no significant difference in expression for genes associated with shared loops (Supplementary Fig. 1f). Consistently, differentially expressed genes (DEGs) between the three cell types are significantly associated with cell-type-specific loops (Supplementary Fig. 1g). Ingenuity Pathway Analysis (IPA)[36] revealed that DEGs between HeLa and K562 categorized based on loop association are enriched in distinct canonical pathways (Fig. 1e). Similar results were obtained in pairwise comparisons between GM12878 and the other two cell lines (Supplementary Fig. 1h, i). For instance, Fig. 1f illustrates the loop architecture and epigenomic features of *ROR2*, which encodes a receptor involved in non-canonical Wnt signaling with a significant role in human carcinogenesis[37,38]. *ROR2* is highly expressed in K562 compared to HeLa, and these CTCF-mediated loops are present only in K562. The up-regulation of *ROR2* expression is associated with a concomitant decrease of H3K27me3 and

increase in H3K36me3 in the region, as well as the appearance of a K562-specific SE in the gene body.

Altogether, cell-type-specific CTCF-mediated loops are prevalent and may play a significant role in the transcriptional programs of cell-type-specific genes. Therefore, we sought to develop a computational approach to infer the CTCF-mediated loops.

**An ensemble learning method to predict CTCF-mediated loops.** We employed a random forest classifier, a tree-based

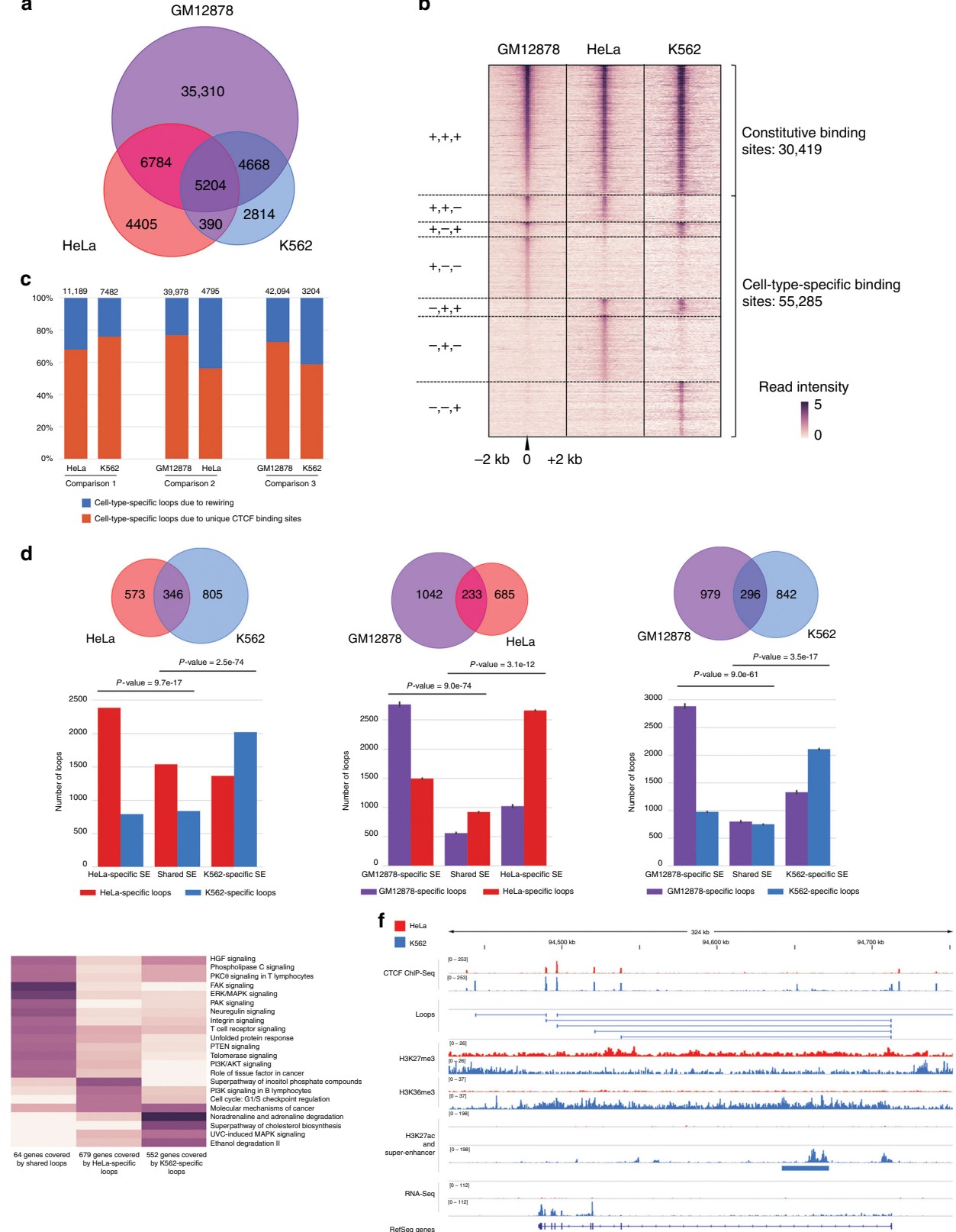

**Fig. 1** CTCF-mediated loops exhibit cell-type-specificity. **a** Venn diagram of CTCF-mediated loops identified from ChIA-PET experiments in GM12878, HeLa, and K562. **b** Heat map of CTCF-binding sites in GM12878, HeLa, and K562 cells. Each row represents a CTCF-binding event identified in ChIA-PET in at least one cell type. The binding sites are divided into seven groups based on the presence (+) or absence (−) of CTCF binding. Color shows the log2-transformed value of reads per kilobase per million reads (RPKM). **c** Cell-type-specific CTCF binding and rewiring between common CTCF-binding sites contribute to cell-type-specific loops. **d** Cell-type-specific Super-Enhancers (SEs) are enriched with cell-type-specific loops. Top: Venn diagram of SEs in pairwise comparison of cell types. Bottom: Number of cell-type-specific loops associated with cell-type-specific and shared SEs. *P*-values were calculated by Chi-square test. The GM12878 ChIA-PET data set was down-sampled to 15% of the original size so that the number of identified loops matched those of the other ChIA-PET data sets. The down sampling and further analysis was repeated 10 times and the 95% confidence intervals were shown. **e** Canonical pathway enrichment analysis of differentially expressed genes associated with K562-specific, HeLa-specific, and shared CTCF-mediated loops, respectively. Color bar represents −log10 (*P*-value). **f** Genome browser snapshot of *ROR2* locus. *ROR2* is expressed and associated with CTCF-mediated loops in K562 but not in HeLa. Expression of *ROR2* in K562 is associated with a concomitant decrease of H3K27me3 and increase of H3K36me3 within the gene body, as well as the appearance of a K562-specific SE. The ChIP-Seq and RNA-seq signals are represented in RPKM values

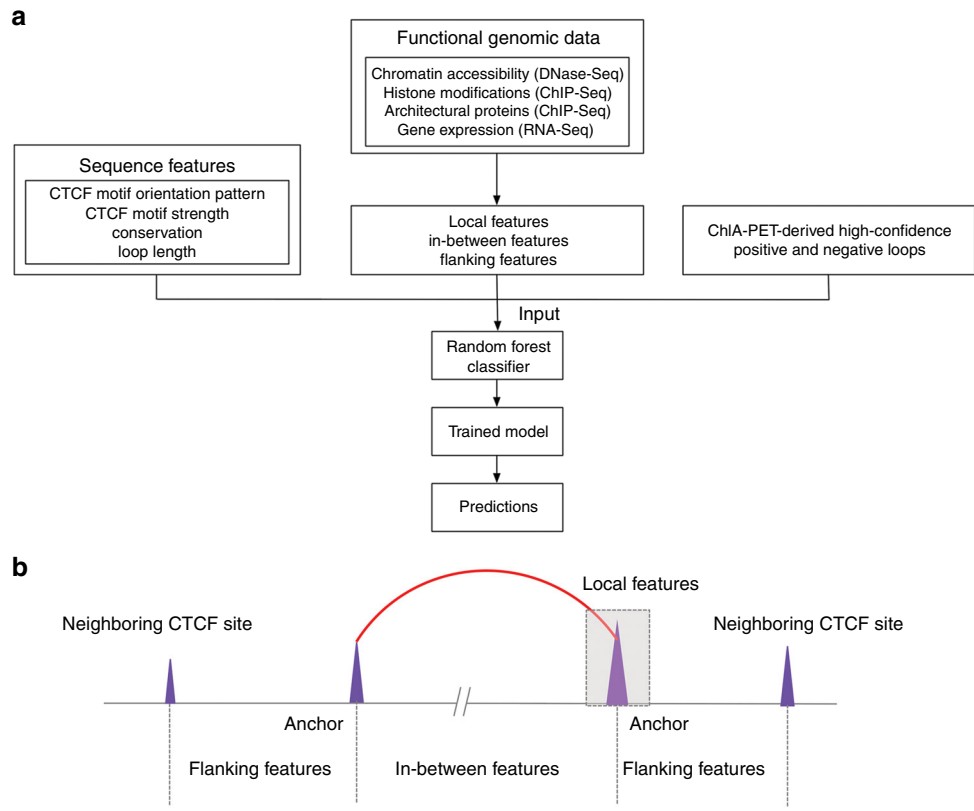

**Fig. 2** Illustration of the Lollipop pipeline. **a** Schematic of the Lollipop pipeline. In training data, positive loops were generated from high-confidence interactions identified from ChIA-PET, and negative loops were random pairs of CTCF-binding sites without interactions in ChIA-PET or significant contact in Hi-C data set. A diverse set of features, generated from genomic and epigenomic data, was used to characterize the interactions. A random forests classifier distinguished interacting CTCF-binding sites from non-interacting ones. The performance of resulting classifier was then evaluated. Trained model can be used to scan the genome and predict de novo CTCF-mediated loops in the same or a different cell type. **b** Illustration of local, in-between, and flanking features

ensemble learning method, to predict CTCF-mediated loops. This classification method takes into consideration the complex interactions among features and is robust against overfitting[39–41]. The pipeline, named Lollipop, aims to find an optimized combination of genomic and epigenomic features to distinguish interacting from non-interacting pairs of CTCF sites. The schema of the pipeline is shown in Fig. 2a. The trained model can be used to predict CTCF-mediated loops in the same or a different cell type.

For training purposes, the positive and negative loops were derived from ChIA-PET data sets[10,29]. To ensure confident labeling of positive loops, we used stringent criteria (FDR ≤ 0.05 and at least two PETs connecting the two anchors). Negative loops were constructed by random pairing of CTCF binding sites and were five times as abundant as the positive loops. Additional rules to select negative loops included: (a) lack of PET in the ChIA-PET data set and (b) absence in the list of identified interactions from the Hi-C experiments (see methods for details).

A total of 77 features were derived from genomic and epigenomic data sets (Fig. 2a). Genomic features include loop length and features defined at the CTCF-binding sites, including CTCF motif orientation, strength, and sequence conservation. We included loop length because it is an inherent determinant of contact frequency between two genomic regions[42], and motif orientation pattern because CTCF anchors preferentially adopt a convergent motif orientation[9]. Epigenomic features include

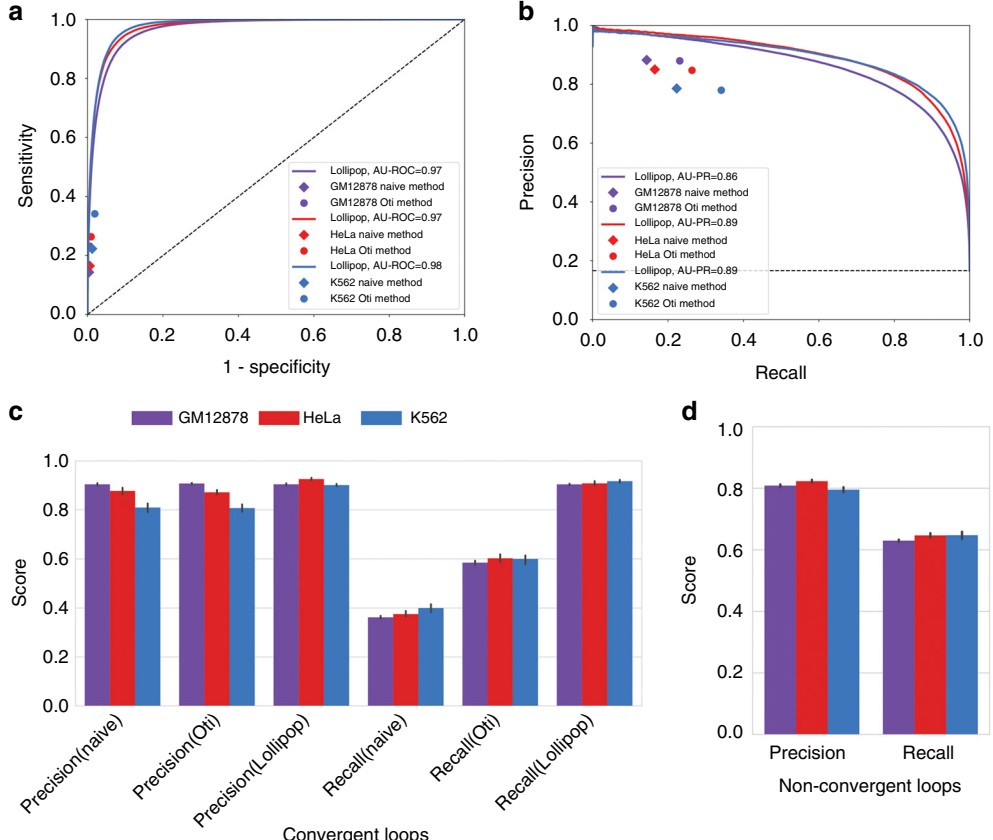

**Fig. 3** Performance evaluation within individual cell types. **a**, **b** Performance evaluation using **a** receiver operating characteristic (ROC) and **b** precision–recall (PR) curve. Performance of the naïve and Oti methods are represented by diamonds and circles, respectively. Results in GM12878, HeLa, and K562 are shown in purple, red, and light blue, respectively. **c** Comparison of the precision and the recall of the three methods in predicting convergent loops. **d** Evaluation of Lollipop's performance on non-convergent loops, which include tandem loops, divergent loops, and loops without CTCF motifs in the anchors. The bars in **c** and **d** denote 95% confidence intervals from 10-fold cross-validation

chromatin accessibility, a variety of histone modifications, and architectural proteins CTCF and Cohesin (RAD21). For the use of DNase-seq and ChIP-seq data sets, three types of features were used: (a) local features defined at the anchors, (b) in-between features defined over the loop region, and (c) flanking features defined over the region from the loop anchor to the nearest CTCF-binding event outside the loop (Fig. 2b). The use of the in-between features was motivated by a recent study[43] showing that signals over the loop regions were more important in predicting promoter–enhancer interactions than signals at anchors. In addition, given the insulator role of CTCF, we reasoned that the potential imbalance of signal intensities on the two sides of CTCF anchors might contribute to the prediction. Therefore, we used the DNase and ChIP-seq signals on the flanking regions as features. Finally, we also included gene expression within the looped region (see Methods for details).

**Performance assessment within individual cell types**. We employed receiver operator characteristic (ROC) and precision–recall (PR) curves with 10-fold cross-validation to assess the performance of Lollipop. To account for possible bias introduced by random partitioning of training data, we performed five iterations for cross-validation and reported the mean performance. For evaluation of Lollipop's performance, two methods were used for comparison. Both methods are inspired by the finding that the CTCF motifs in anchors preferentially adopt convergent orientation:[9,10] (a) The naïve method, which pairs a CTCF-bound motif that resides on the forward strand to the

nearest downstream CTCF-bound motif that resides on the reverse strand (Supplementary Fig. 2a); (b) The Oti method[44], which iteratively applies the naïve method to CTCF-binding sites selected by different signal intensity thresholds (see Supplementary Fig. 2b for illustration and Methods for details). By doing so, the Oti method identifies more loops than the naïve method and partially recovers the nested structure of some CTCF-mediated loops.

Figure 3a, b shows that Lollipop achieved an area under ROC curve (AU-ROC) value of ≥0.97 and area under PR curve (AU-PR) value of ≥0.86 in all cell lines. Compared to other methods, Lollipop achieved similar or higher precision and superior recall. The latter can be partially attributed to the failure of naïve and Oti methods to capture tandem loops or loops without CTCF motif on anchors, which account for a significant fraction of CTCF-mediated loops (64% for GM12878, 61% for HeLa, 49% for K562). We then independently evaluated Lollipop's performance on convergent and non-convergent loops. Even on convergent loops, Lollipop achieved a superior recall score with a precision score comparable to those of the naïve and Oti methods (Fig. 3c). Furthermore, Lollipop also performed well in predicting non-convergent loops (Fig. 3d). In summary, Lollipop can account for the complexity of loop structures by integrating genomic and epigenomic features and outperforms methods that only consider the convergent CTCF motif orientation.

**Identification of determinants of CTCF-mediated loops**. Considering that convergent motif orientation does not suffice to

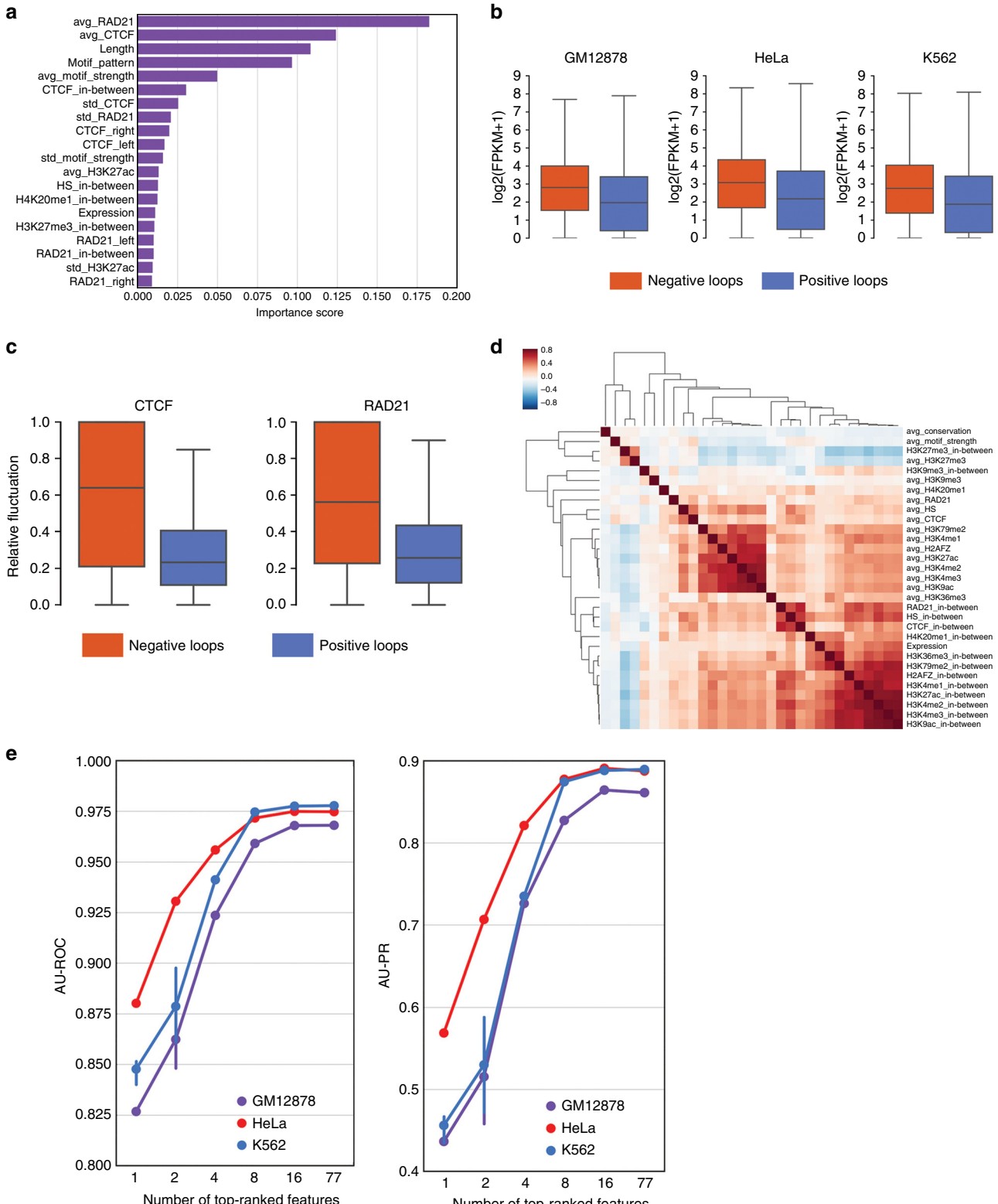

identify CTCF-mediated loops, we ranked features that significantly improve the performance, by measuring the mean decrease in impurity during training the random forests classifier[45]. We found that the average binding intensity of CTCF and Cohesin (RAD21) at the loop anchors are the most important features (Fig. 4a and Supplementary Fig. 3a), suggesting that sites with stronger CTCF and Cohesin binding are more likely to become anchors (Supplementary Fig. 3b), and consistent with the observation that that these proteins are important for chromatin interactions[14,15]. In agreement with previous results, loop length and motif orientation pattern were among the top features[9,42]. The list also includes features defined within loop regions, among which gene expression was of particular interest. Regions inside positive loops exhibit significantly lower gene expression levels

**Fig. 4** Identification of features with predictive power. **a** Ranking of predictive importance of the top 20 features in the model trained in GM12878 cells. Predictive importance is measured by mean decrease impurity in the training process. "avg" and "std" represent the mean and standard deviation of the signal intensity on both anchors. "_left" and "_right" represent the flanking features while "_in-between" is the signal intensity within the loop. **b** Distributions of average gene expression levels within negative and positive loops. The positive and negative loops were defined in the training data, with loops not containing promoters excluded from the analysis. Center lines, boxes, and whiskers of the box plot represent the median value, first/third quartiles, and 1.5 interquartile range of the samples, respectively (same below). $P$-value <1e-300 for GM12878 and HeLa, $P$-value = 1.3e-271 for K562, Mann–Whitney $U$-test. **c** Distribution of the relative fluctuations of CTCF and RAD21 binding intensities on paired anchors of negative and positive loops in GM12878 cells. Relative fluctuation was defined as the ratio of standard deviation to mean intensity of anchor pairs. In both cases, $P$-value <1e-300, Mann–Whitney $U$-test. **d** Heat map of feature correlations in GM12878. On anchors, active histone marks are highly correlated. Along the loop regions, active histone marks and expression exhibit strong correlation. In addition, RAD21, CTCF, and DNase hypersensitive sites are strongly correlated. Spearman's rank correlation and hierarchical clustering were used. **e** Recursive Feature Elimination analysis on feature reduction. Left: AU-ROC; right: AU-PR. Bars represent the 95% confidence intervals from five runs

compared to negative loops (Fig. 4b). This finding is supported by similar trends exhibited by histone marks for active gene bodies H3K79me2 and H3K36me3 (Supplementary Fig. 3c). Another interesting feature is the standard deviation of CTCF and Cohesin binding at the anchors (Fig. 4a). We therefore examined the relative fluctuation, defined as standard deviation divided by average intensity, of CTCF and Cohesin on anchor pairs of the positive and negative loops. As shown in Fig. 4c and Supplementary Fig. 3d, anchor-pair CTCF and RAD21 have significantly lower relative fluctuation in positive loops than in negative loops.

While CTCF binding at anchors is clearly critical for looping, formation of a loop requires wiring (i.e., physical interaction) between specific pair of anchors. We therefore asked what features contribute to the wiring. To this end, we considered negative loops to be random pairings of actual anchors, and then reanalyzed feature importance. As shown in Supplementary Fig. 3e, length, motif orientation, and expression are strongly contributing, whereas CTCF and Cohesin binding at anchors become much less important. It is worth noting that more in-between features showed up on the list, compared to those in Fig. 4a and Supplementary Fig. 3a.

As the features employed are correlated (Fig. 4d and Supplementary Fig. 3f), the feature importance scores might be skewed. To validate the ranking of feature importance, we applied the Recursive Feature Elimination (RFE) method to evaluate the performance of the recursively reduced feature set. The results are consistent with the feature ranking from the mean decrease impurity (Supplementary Table 3). Last, performance evaluation under different feature sets suggests that near-optimal performance can be achieved by using ~16 features (Fig. 4e). These features include those derived from CTCF and RAD21 binding, loop length, CTCF motif orientation, gene expression, as well as epigenetic features (Supplementary Table 3). Of note, CTCF-binding intensity and motif contain non-redundant information, and Lollipop performs reasonably well when features derived from CTCF and Cohesin ChIP-seq data are not available (Supplementary Fig. 3g).

**Performance assessment across cell types**. Having demonstrated Lollipop's superior performance within individual cell types, we next used the model trained in one cell type to make predictions and assessment in another cell type (see Methods for details). This is more realistic and challenging, as a large number of CTCF-mediated loops are cell-type-specific. In all three cell types Lollipop achieved AU-PR ≥ 0.80 and AU-ROC ≥ 0.94 (Fig. 5a and Supplementary Fig. 4a), only moderately lower than its performances within individual cell types (Fig. 3a, b) and outperforming motif-orientation-based methods (Fig. 5a and Supplementary Fig. 4a). Furthermore, when evaluated on cell-type-specific loops, Lollipop achieved slightly lower performance (AU-PR ≥ 0.72 and AU-ROC ≥ 0.93) (Fig. 5b and Supplementary

Fig. 4b), suggesting that Lollipop predicts cell-type-specific loops reasonably well.

Given that a loop consists of a pair of anchors and the wiring between them, we then dissected Lollipop's predictive power on anchors and wiring, respectively. For assessment of anchor prediction, we evaluated Lollipop by comparing the anchor usage of the predicted loops with that of loops identified from ChIA-PET in the target cell type. For assessment of wiring prediction, we constructed negative loops by random pairing of actual anchors in the target cell type (see Methods for details). Figure 5c, d show the PR curves demonstrating that Lollipop performed reasonably well in both, and better in predicting anchors than in predicting wiring. The results of ROC (Supplementary Fig. 4c, d) are consistent with those of PR.

**Evaluation of de novo predictions of CTCF-mediated loops**. After training Lollipop in individual cell types, we then applied it to scan the genome of the same cell type to make de novo genome-wide predictions. Lollipop predicted 67,920, 38,688, and 32,430 loops in GM12878, HeLa, and K562, respectively. Notably, the number of predicted loops in GM12878 is much larger than those of the other two cell types, due to the much larger number of loops identified by ChIA-PET in GM12878 (see the last column of Supplementary Table 2). These loops were used in training the model and thus affect the number of predicted loops. Indeed, if we down-sample the GM12878 ChIA-PET library to 15% so that the number of called loops is on par with those in K562 and HeLa (Supplementary Table 2), the number of predicted loops is comparable to the number of predictions in K562 and HeLa.

The predicted loops can be classified into three categories according to the levels of support from ChIA-PET data (see Methods for details): "Significant" (FDR ≤ 0.05 and PET number ≥ 2), "With evidence" (FDR > 0.05 or PET number = 1), and "No support" (PET number = 0). If the prediction stringency is increased, higher percentage of predicted loops finds support in ChIA-PET data and higher percentage falls into the "Significant" category (Supplementary Fig. 5a). Moreover, loops in the "Significant" and "With evidence" categories are more likely to have convergent motifs (Supplementary Fig. 5b).

As shown in Supplementary Fig. 5c, a large fraction of the predicted loops (48%, 73%, and 77% for GM12878, HeLa, and K562, respectively) were not supported by ChIA-PET under the stringent criterion of FDR ≤ 0.05 and PET ≥ 2 used for defining positive loops. However, if we relaxed the stringency to PET number ≥1 in ChIA-PET, the fraction of predicted loops not supported by ChIA-PET was significantly reduced to 24%, 42%, and 50% in GM12878, HeLa, and K562, respectively. Similar result can be obtained with the down-sampled GM12878 library (Supplementary Fig. 5d). This observation raises the question of whether the predicted loops with less or no ChIA-PET support

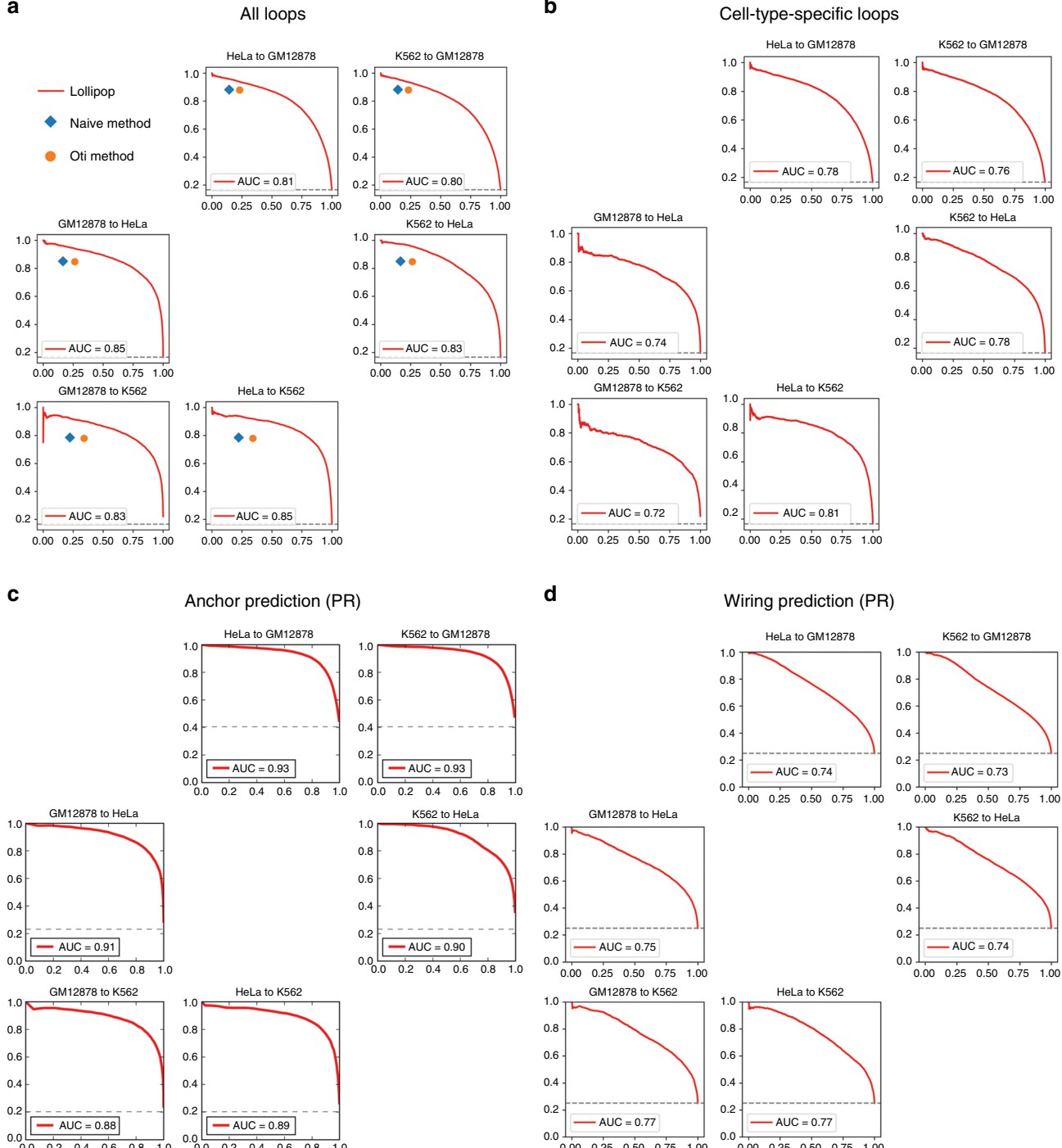

**Fig. 5** Performance evaluation of Lollipop across cell types. **a** Across-cell-type performance evaluation using PR curves. In each subplot, "cell A to cell B" applies the model trained from cell-type A to the data of cell-type B. For comparison, the performance of the naïve and Oti methods in each cell type were represented by diamonds and circles, respectively. **b** Across-cell-type performance evaluation on cell-type-specific loops using PR curves. **c** Performance evaluation of anchor prediction using PR curve. **d** Performance evaluation of wiring prediction using PR curve. The dash lines in **a**–**d** represent baseline performance

are indeed false positives. To address this question, we carried out the following computational and experimental evaluations on those predicted loops without any ChIA-PET support.

First, we used the published Hi-C contact matrices for GM12878 and K562 (ref. [9]) (see Methods for details) to evaluate the loops in the "No support" category, and found that they have significantly higher contact frequencies than pairs of randomly

chosen genomic loci (Fig. 6a). For fair comparison, the control regions were sampled to have a length distribution matching those of the target loops. Notably, loops in the "With evidence" and "Significant" categories are also supported by Hi-C data and exhibit higher contact frequencies than the loops in the "No support" category (Supplementary Fig. 5e). Second, we randomly selected two such cases and performed 3C experiments

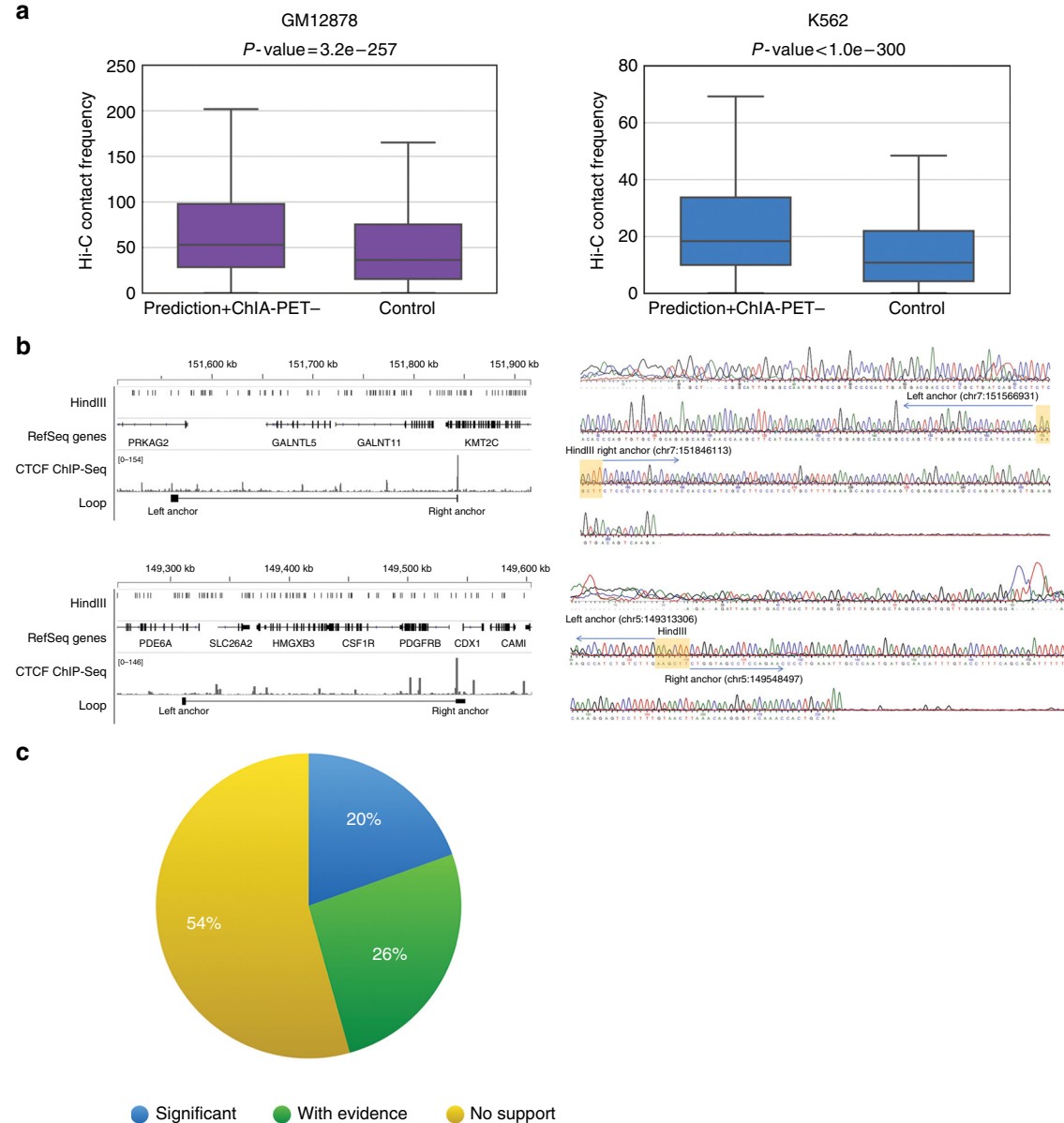

**Fig. 6** Validation of predicted CTCF-mediated interactions. **a** CTCF-mediated loops predicted by Lollipop but lacking ChIA-PET support exhibit significantly higher contact frequency than background in Hi-C experiments. *P*-values were calculated using Mann–Whitney *U*-test. **b** Validation of two loops predicted by Lollipop, but not present in the HeLa ChIA-PET data set. Left: schematic of PRKAG2-KMT2C (chr7: 151560677–151843260; top) and PDE6A-PDGFRB loop (chr5: 149312517–149547724; bottom). Right: Sanger sequencing confirmation of the ligation junctions. Shaded areas in the right panels indicate the HindIII ligation junctions. **c** Scaling analysis of loop prediction. Loops predicted using a model trained on the down-sampled (to 15%) GM12878 data, but lacks support in the down-sampled data (i.e., the yellow slice in Supplementary Fig. 5d) are evaluated by the full ChIA-PET data. Forty-six percent of these loops find support

(Supplementary Fig. 5f). Figure 6b shows the sequence of the ligation junctions from the long-range interactions (*PRKAG2-KMT2C* and *PDE6A-PDGFRB*) in HeLa. 3C-qPCR further confirmed the contact frequency of the PRKAG2-KMT2C loop in respect to neighboring HindIII fragments (Supplementary Fig. 5g).

Having shown that the predicted loops lacking ChIA-PET support could be real, we sought to understand why they were not observed in ChIA-PET. To this end, we performed scaling analysis in the ChIA-PET data of GM12878 cells, which received higher sequencing coverage compared to K562 and HeLa cells (Supplementary Table 2). Specifically, we used the 15% down-sampled GM12878 ChIA-PET library to identify loops with the same approach employed for the full data set, and trained a

classifier. We then applied this classifier to make genome-wide predictions. Of the 33,206 predicted loops, 11,954 are without any support from the down-sampled ChIA-PET data set. However, 46% of these loops find support in the full ChIA-PET library, and 20% of these loops even find significant support (Fig. 6c). Taken together, the scaling analysis suggests that insufficient sequencing depth contributes to the presence of predicted loops lacking support in ChIA-PET.

To evaluate the robustness of the model prediction against training data, we compared the genome-wide predictions in GM12878 derived from the HeLa and K562 models. The looping probabilities of all potential loops in GM12878 predicted from HeLa and K562 models are concordant (Supplementary Fig. 5h). Moreover, the predicted loops from HeLa and K562 models

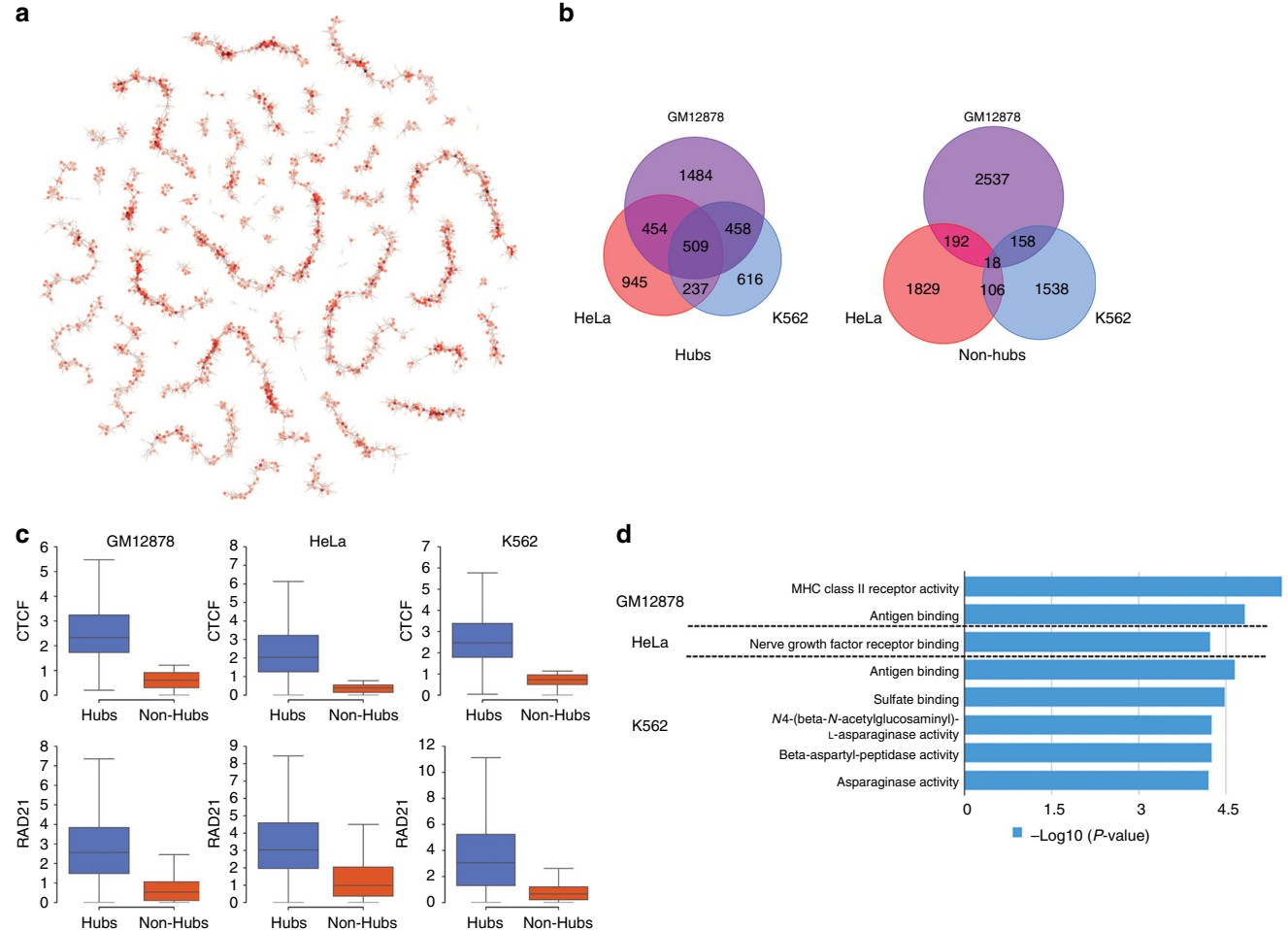

**Fig. 7** Topological properties of the CTCF-mediated interaction network and their association with biological functions. **a** Visualization of the CTCF-mediated interaction network of chromosome 1 in GM12878 cells. Each node represents an anchor, with color representing the degrees of connection. Each edge represents an interaction. **b** Overlap of predicted hubs and non-hubs among each cell type. Hubs are more conserved than non-hubs. **c** Distribution of the binding affinity of architectural proteins, CTCF (top) and RAD21 (bottom), on predicted hubs and non-hubs. *P*-values <1e-300 for all cases, Mann–Whitney *U*-test. **d** Functional enrichment analysis of hubs using GREAT. The *x*-axis represents the binomial *P*-values

significantly overlap (Supplementary Fig. 5i). Taken together, genome-wide de novo predictions from models trained from different cell types are congruent.

**Topological properties of CTCF-mediated interaction network**. To better understand these interactions, we took a systems approach to visualize and analyze the CTCF-mediated interactions. We constructed the CTCF-mediated interaction network by denoting the anchors as nodes and the long-range interactions as edges. As exemplified in Fig. 7a, where the interaction network on chromosome 1 (visualized using Graph-tool) is shown, the CTCF-mediated interactions form a disconnected network encompassing many linear-polymer-like components. This is dramatically different from the RNA-PolII-mediated interaction network[46], which is dominated by one scale-free connected graph[46]. This dramatic difference in topological structure is also manifested in the degree distributions (Supplementary Fig. 6), where the distribution for RNA PolII exhibits a fatter tail.

It is worth noting that the degrees of connection among the anchors vary. We therefore examined CTCF hubs, anchors involved in multiple interactions. Ranking anchors according to the degrees of connection, we defined hubs as those among the top 10% anchors and non-hubs as the bottom 10% (see Methods

for details), and identified 2905, 2145, and 1820 hubs for GM12878, HeLa, and K562, respectively. Subsequent comparison between hubs and non-hub nodes revealed that hubs are (a) more conserved across cell types than non-hubs (Fig. 7b), likely because they serve as the structural foci of genome organization in the nucleus; (b) characterized by significantly higher binding affinity for CTCF and Cohesin (Fig. 7c); and (c) associated with distinct biological functions. Gene ontology analysis[35] showed that the hubs are preferentially associated with immunology-related functions in GM12878 and K562 cells, but not in HeLa cells (Fig. 7d), consistent with the cellular origin of these cell lines. For example, the hubs in GM12878 and K562 cells that are of hematopoietic origin were significantly associated with antigen binding, and the GM12878 hubs were significantly associated with the MHC (major histocompatibility complex) and MHC class II (MHC-II) genes, which are essential for immune system. This result is consistent with an important role for CTCF in controlling MHC-II gene expression[47,48]. To explore the mechanistic basis of the enrichment of cell-type-specific functions of hubs, we examined the enrichment of transcription factor motifs at the hubs using SeqPos[49]. We found that the enrichment pattern is cell-type-specific (Supplementary Fig. 6b), suggesting CTCF hubs may be associated with distinct sets of transcription factors in different cell types. Notably, ZFX, ARNT, and MZF1

are only enriched in GM12878 and K562 cells, consistent with their roles in the development of hematopoietic system[50–52].

## Discussion

Here we showed that CTCF-mediated chromatin interactions exhibit extensive variations across cell types. These cell-type-specific interactions are functionally important, as they are linked to DEGs and cell-type-specific SEs contributing to cell identity. However, genome-wide profiling of CTCF-mediated interactions is available in a very limited number of cell types and conditions, as experimental approaches remain challenging and costly. Therefore, we developed Lollipop, a machine-learning framework, to make genome-wide predictions of CTCF-mediated loops using widely accessible genomic and epigenomic features. Using computational as well as experimental validations, we demonstrated that Lollipop performed well within and across cell types. Analysis of the machine-learning model revealed features associated with CTCF-mediated loops, and shed light on the rules underlying CTCF-mediated chromatin organization.

While previous studies focused on the significance of conserved CTCF binding at TAD boundaries or loop anchors, our study showed a significant proportion of CTCF-mediated interactions are cell-type-specific. Based on our analysis, both lineage-specific recruitment of architectural proteins and alternative wiring among available anchor sites contribute to the establishment of cell-type specificity. Although the process of establishing cell-type-specific loops is not well understood, it is conceivable that multiple factors combine to orchestrate a cell-type-specific chromatin context to promote the formation of a loop.

The convergent orientation of CTCF motifs at loop anchors is a prominent feature of CTCF-mediated interactions[9,10], as it is also manifested by our model. However, model comparison demonstrated that motif orientation alone is limited in its predictive power, and inclusion of other features significantly improved the performance. Interestingly, we found that features for the loop regions, which are away from the anchors, contribute significantly to the predictive power, consistent with findings in enhancer–promoter interaction predictions[43]. Specifically, gene expression exhibits distinct distributions over positive loop regions compared to negative loops (Fig. 4b, and Supplementary Fig. 4c), which may be attributed to the enhancer-blocking role of CTCF loop anchors.

In evaluating our predictions, we showed that false-positive loops could be due to mislabeling in the testing data. As advances in experimental protocols and continuous decreases in sequencing cost would result in better training data in reference cell types, it is likely that the performance of Lollipop would further improve. Since CTCF plays a major role in defining regulatory domains, results obtained from our approach can potentially be used as constraints in predicting enhancer–promoter interactions, which remains a major challenge. Overall, CTCF-mediated chromatin interactions are critical for genome organization and function, and our study provides a computational tool for the exploration of the 3D organization of the genome.

## Methods

**Identification of CTCF-mediated loops from ChIA-PET data**. We employed ChIA-PET2 (v0.9.2)[30] to identify CTCF-mediated loops. Briefly, ChIA-PET2 involves linker filtering, PET mapping, PET classification, binding-site identification, and identification of long-range interactions. In the step of linker filtering, one mismatch was allowed in identifying reads with linkers. After linker removal, only reads with at least 15 bp in length were retained for further analysis for GM12878 and HeLa (read length = 150 bp). For K562, the read length was shorter (36 bp), therefore reads with at least 10 bp in length were retained for further analysis. In other steps, default values for parameters were used. Only uniquely mapped reads were kept, and PETs were de-duplicated. Significant loops were identified with false

discovery rate (FDR) ≤ 0.05 (ref. [53]). We further required that they are supported by at least two PETs.

We only considered long-range interactions whose lengths are less than 1 million bp (mb), for two reasons. First, the vast majority of loops (93.2% for GM12878, 97.3% for HeLa, 98.1% for K562) are less than 1 mb long. Similar observations were previously made[10]. Second, insulated neighborhoods, previously defined as CTCF-mediated loops that are co-bound by cohesin and contain at least one gene, were found to range from 25 to 940 kb (refs. [6,16]) (reviewed in ref. [13]).

**Comparison of CTCF-mediated loops among cell types**. An anchor is considered as shared by two cell types if the respective genomic regions delineating this anchor overlap in the two cell types. The loops shared by all three cell types were defined as GM12878 loops shared by both K562 and HeLa.

**Analysis of CTCF-binding sites in three cell types**. CTCF peaks were determined by MACS2 (ref. [54]) in the ChIA-PET2 pipeline. A binding site was defined as the peak summit ± 500 bp. The binding sites in the three cell types were classified into seven groups according to the overlapping pattern. Binding intensity for each site was represented by the log2 (RPKM) value over the summit + 2 kb region. For each group, the binding sites were ordered in descending order according to binding intensity in a prioritized manner. Namely, CTCF-binding sites present in GM12878 were ordered by their binding strength in GM12878; CTCF-binding sites not present in GM12878 were ordered by binding strength in HeLa and then in K562 accordingly. Seaborn (V0.7.1, [http://seaborn.pydata.org]) was used to generate the heat map.

**SE analysis**. SEs are defined as stretches of chromatin that cluster multiple enhancers decorated with H3K27ac[31]. They were identified by the Ranking Ordering of Super-Enhancers algorithm (ROSE[32,33]), using H3K27ac ChIP-Seq data as input and default parameters. Identified SEs were then uploaded to Genomic Regions Enrichment of Annotations Tool (GREAT) V3.0.0 (ref. [35]) for GO analysis (Supplementary Fig. 1e). If an SE in one cell type does not overlap with any SEs in a different cell type, it is deemed as an SE specific to that cell type. If an SE in one cell type overlaps with an SE in the other cell type by at least 1 bp, it is called a shared SE. We then counted the number of cell-type-specific loops associated with each type of SEs. If a loop overlaps with an SE by at least 1 bp, we considered the loop associated with the SE. The comparison between HeLa and K562 is shown in Fig. 1d. For comparison between GM12878 and another cell type, the 15% down-sampled GM12878 ChIA-PET data set was used so that the number of loops identified matched those from the ChIA-PET data sets of the other two cell types (see Supplementary Table 2). Then analysis identical to that in Fig. 1d was carried out. The down sampling and follow-up analysis were repeated 10 times to ensure reproducibility, and 95% confidence intervals were shown in the Fig. 1d.

**Association of gene expression with CTCF-mediated loops**. Each cell-line has two RNA-Seq replicates. Cufflinks V2.2.1 (ref. [55]) with default parameters (q-value = 0.05) was used to identify the DEGs.

To test the effects of cell-type-specific loops on gene expression (Supplementary Fig. 1f), we identified genes associated with cell-type-specific and shared loops in pairwise comparisons of cell types. A gene is associated with a loop if its promoter region (TSS ± 2 kb) is inside the loop.

For comparison between HeLa and K562 in Supplementary Fig. 1g, a DEG was deemed to be associated with HeLa-specific loops if it is within one or more HeLa-specific loops but not within any K562-specific loops. If a DEG is covered only by one or more shared loops, this DEG is deemed to be associated with shared loops. Following the criteria described above, we obtained three sets of DEGs respectively associating with HeLa-specific loops, shared loops, K562-specific loops. These three sets of DEGs were then subject to GO analysis using "Ingenuity Pathway Analysis"[36]. The GO terms whose P-values are no less than 1e-3 in all three gene sets were then removed. The result is shown in Fig. 1e. Color key represents the −log10 (P-value) value. For comparison between GM12878 and another cell type (Supplementary Fig. 1h, i), the GM12878 ChIA-PET library was first randomly down-sampled to 15% of the original size so that the number of loops identified matched those of the ChIA-PET libraries from the other two cell types. For Supplementary Fig. 1g, non-DEG genes were those with the least significant expression changes as ranked by P-value, with group size matching to that of the corresponding DEG group.

**Identification of CTCF motif occurrences**. The position frequency matrix of CTCF for human was downloaded from Jaspar 2016 ([http://jaspar.genereg.net])[56]. CTCF motif occurrences were identified by the FIMO package (V4.11.1 (ref. [57])) with the P-value <1e-5. In total, 110,879 motif occurrences were identified.

**Preparation of training data**. Positive loops were identified using the ChIA-PET2 pipeline with FDR ≤ 0.05 and IAB ≥ 2, with loop length restricted to be in the range of 10 kb to 1 mb. The choice of the lower limit of 10 kb is because the ChIA-PET-identified loops with length below 10 kb are likely caused by self-ligation in library preparation[25]. The reason for the upper limit of 1 mb was given above. Negative

loops were constructed by random pairing of CTCF-binding sites, with loop length ranging from 10 kb to 1 mb. The number of negative interactions was chosen to be five times that of the positive interactions. To ensure accurate labeling, we further required that the negative loops (1) do not receive any ChIA-PET support and (2) are not present in the CTCF-mediated interactions identified from the Hi-C experiments[9].

**Feature calculation**. Genomic features include motif strength, motif orientation, conservation score, and loop length. Motif strength represents how similar the underlying sequence is to the CTCF consensus motif. The motif strength score was provided by FIMO[57]. The motif strength score of a CTCF-binding site (summit ± 1000 bp) was represented by the strength of the motif occurrence within the site. If a CTCF-binding site has more than one motif occurrences, the highest score was used. If there is no motif occurrence, 0 would be assigned. The feature of motif orientation was represented by the following rule: If neither anchor has CTCF motif, we assigned a value of 0; If one anchor has no motif and the other has one or more than one motifs, we assigned a value of 1; if both anchors have one or more motif occurrences, the orientation of each anchor was determined by the orientation of its strongest motif occurrence. Divergent orientation would be assigned a value of 2, tandem orientation a value of 3, and convergent orientation a value of 4. For conservation, we used the 100 way phastCons score downloaded from UCSC ([http://hgdownload.cse.ucsc.edu/goldenpath/hg19/phastCons100way])[58]. The conservation score of a CTCF-binding site was defined as the mean value of the conservation score of each nucleotide in the summit ± 20 bp region.

Functional genomic features include chromosome accessibility profiled by DNase-Seq, histone modifications, CTCF and Cohesin binding profiles profiled by ChIP-Seq, and gene expression profiled by RNA-Seq. DNase-Seq and ChIP-Seq data were de-duplicated and then subject to pre-processing to remove noise as follows. For DNase-Seq data, peaks were downloaded from ENCODE[29]. For ChIP-Seq data, SICER (V1.1)[59] was used to identify enriched regions with FDR 1e-5. For histone modifications with diffused signal (H3K27me3, H3K36me3, H3K9me3, H3K79me2), window size = 200 bp, gap size = 600 bp were used. For other ChIP-Seq libraries, window size = gap size = 200 bp were used. For both DNase-Seq and ChIP-Seq, only reads located on signal-enriched regions were used for feature calculation. For RNA-Seq data, gene expressions were calculated using Cufflinks[55] with default parameters. Each data set was characterized by three types of features: local features, in-between features, and flanking features, as illustrated in Fig. 2b. Local features are defined around anchors, represented by the signal intensity (RPKM value) over the CTCF summit position ± 2 kb region. In-between features are represented by the average signal intensity (RPKM value) over a presumed loop region. The value of the expression feature is defined as the average FPKM value of the genes whose promoters are located inside the presumed loop (i.e., total expression divided by the number of genes). The flanking features are represented by the RPKM value over the region from the loop anchor to the nearest CTCF-binding event identified in the CTCF ChIP-Seq.

**Implementation of the naïve method and the Oti method**. The naïve method is implemented by pairing a CTCF-bound motif that resides on the forward strand to the nearest downstream CTCF-bound motif that resides on the reverse strand (Supplementary Fig. 2a). The Oti method was introduced in ref. [44]. It ranked all the active motif sites in terms of CTCF peak strength in descending order. First, all active motif sites were used to construct loops by the naïve method. Then, the same procedure was repeated for the top 80%, top 60%, top 40%, and top 20% active motif sites. The loops constructed in different rounds were then pooled together. The Oti method is illustrated in Supplementary Fig. 2b.

**Performance evaluation within individual cell types**. In Fig. 3c, d, the performance was evaluated at the looping probability cut-off of 0.5.

**Evaluation of feature importance**. Predictive importance scores of features were obtained from the attribute of "feature_importances" of the trained random forest classifier[60]. After ranking, the top 20 features were visualized in Fig. 4a and Supplementary Fig. 3a. The correlation matrix was subject to hierarchical clustering, as shown in Fig. 4d and Supplementary Fig. 3f. RFE method was used to validate the analysis of the feature importance. After each iteration, model performance was evaluated in terms of AU-ROC curve and AU-PR curve. The performance vs feature number was plotted in Fig. 4e.

For feature importance analysis of wiring prediction (Supplementary Fig. 3e), negative data were prepared as follows: the anchors of positive loops were used to construct negative loops by random pairing. The number of negative loops was set to be three times that of positive loops. Other procedures on construction of negative loops were the same as described in the section of "Preparation of training data". Positive data remained unchanged.

**Performance evaluation across cell types**. In the across-cell-type performance evaluation, the model trained in cell-type A was applied to cell-type B, using training data prepared in B for evaluation of performance.

In the across-cell-type performance evaluation of cell-type-specific loops, the positive loops used for evaluation are loops specific to the target cell type. The

negative loops are random pairings of CTCF-binding sites and are five times as abundant as the positive loops (Fig. 5b and Supplementary Fig. 4b).

For evaluation of anchor prediction, the anchors of positive loops in cell-type B were labeled positive, while the anchors belonging only to negative loops in cell-type B were labeled negative. The anchors of predicted loops were compared with positive and negative labels for evaluation of anchor prediction. This evaluation was repeated under different thresholds of looping probability to generate the PR and ROC curves (Fig. 5c and Supplementary Fig. 4c).

For evaluation of wiring prediction, the anchors of positive loops in cell-type B were used to construct negative loops by random pairing. The model trained in cell-type A was then applied to the training data of cell-type B for evaluation (Fig. 5d and Supplementary Fig. 4d).

**Computational evaluation of predicted CTCF-mediated loops**. A model trained in a cell type was used to predict loops genome-wide in the same cell type. Predicted loops were then compared with loops identified from ChIA-PET data sets and categorized into three groups. "Significant" loops denote those supported by ChIA-PET under the stringent criterion of FDR ≤ 0.05 and PET ≥ 2. "With evidence" loops denote those supported by ChIA-PET reads but do not meet the stringent criterion mentioned above. "No support" loops denote those without any support from ChIA-PET (Supplementary Fig. 5a, c).

For down sampling of ChIA-PET library in GM12878 cells, the ChIA-PET library was first randomly down-sampled to 15% of the original size, followed by loop identification using ChIA-PET2 and preparation of training data. Trained model was used to make genome-wide predictions. The predicted loops were categorized into three groups by comparing with loop calls using the down-sampled library, as described above. The result was shown in Supplementary Fig. 5d.

For evaluation of predicted loops using Hi-C data (Fig. 6a, Supplementary Fig. 5e), 10 kb resolution Hi-C contact matrices for GM12878 and K562 (ref. [9]) were used for validation. The contact matrices were normalized by Knight and Ruiz (KR) normalization vector[9]. For each cell type, we collected contact frequencies from the contact matrix for the predicted loops. As a control, we chose a set of random pairings of genomic locations as anchors with matching size and length distribution. We then collected the contact frequencies of this control set. The two contrasting distributions of contact frequencies are shown. HeLa cell was not included in this analysis because the Hi-C library and Hi-C derived contact matrix are not available.

For scaling analysis in GM12878 cells, predicted loops belonging to the "No support" group in the down-sampled ChIA-PET library (yellow slice in Supplementary Fig. 5d) were compared with the loops identified using the full GM12878 ChIA-PET library and categorized into three groups, as shown in Fig. 6c.

**Chromosome Conformation Capture**. The loops used for experimental validation were randomly selected from the loops predicted by Lollipop but not observed in ChIA-PET in HeLa cells, as described above. HeLa cells were purchased from ATCC (#CCL-2). They were tested for mycoplasma and were found negative. For the 3C assay, cells were fixed and nuclei were prepared as in ref. [61]. Nuclei were permeabilized with SDS, and subsequently DNA was digested overnight with HindIII in situ. The next day, the samples were diluted 10-fold in T4 ligation buffer and proximity ligation took place at 16 °C for 4 h and continued at room temperature for 45 min. Reverse crosslinking was performed overnight by Proteinase K treatment. Next, samples were treated with RNase A for 1 h, and 3C DNA library was extracted and purified using phenol–chloroform. The digestion efficiency, as well as the quality and quantity of 3C libraries, was assessed before downstream analyses. The Q5 Taq polymerase (NEB) was used for PCR reactions using the following protocol: 98 °C 30 s, 35 cycles [98 °C 10 s, 70 °C 15 s, 72 °C 10 s], 72 °C 2 min. Reactions were run on 2% agarose gels and analyzed using the ImageLab software (BioRad). Bands were extracted and sequenced (Eurofins) to confirm specificity of primers and loop identity. Data points plotted are the averages of duplicates ± SD from two independent library preparations. Primers (*KMT2C*: U2, U1, L, D1, D3, D4, R (from upstream to downstream); and *PDGFRB*: L, R) were designed using a uni-directional strategy[62] and sequences are provided in Supplementary Table 4.

**Analysis of CTCF-mediated interaction network**. To construct CTCF-mediated interaction network, we used nodes to represent anchors and edges to represent loops. Graph-tool (V2.22 [https://graph-tool.skewed.de]) was used for visualization of networks (Fig. 7a). In identification of hubs, anchors were ranked according to the degrees of connection in descending order. Anchors with the same degrees of connection were further ranked according to CTCF-binding intensity in descending order. The top 10% anchors were defined as hubs, while the bottom 10% as non-hubs.

For functional enrichment analysis of hubs (Fig. 7d), hubs were uploaded to GREAT (V3.0.0)[35] for functional enrichment analysis. The whole set of CTCF anchors was used as background. The GO terms in "Molecular Functions" with P-value <1e-4 in each cell type were shown.

**Code availability**. Lollipop is publicly available in [https://github.com/ykai16/Lollipop].

## Data availability

GM12878 and HeLa ChIA-PET data were downloaded from the Gene Expression Omnibus (GEO) with accession number GSE72816 (ref. [10]). K562 ChIA-PET data were downloaded from ENCODE[29] with accession number ENCLB559JAA. High-resolution genome-wide Hi-C contact matrices were obtained from GEO with accession number GEO63525 (ref. [9]). DNase-Seq, ChIP-Seq, and RNA-Seq data were downloaded from ENCODE and were aligned to hg19. The accession numbers for the data used in this study were summarized in Supplementary Table 1.

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

## Acknowledgements

We thank Dr. Michael Beer for valuable discussions and suggestions. We thank Nick Waring, Stephanie Perkail, and Coen Lap for proof-reading and editing. This research was supported by National Institute of Health (Division of Intramural Research of National Lung and Blood Institute) to J.Z., National Cancer Institute grants (R00CA158582, R21CA182662, and R03CA212068) to A.T., a George Washington University Cross-Disciplinary Research Fund to A.T. and W.P., and National Institute of Allergy and Infection Diseases grants (R21 AI113806, R01 AI121080) to W.P.

## Author contributions

Y.K., A.T., and W.P. conceived the project. A.T and W.P. supervised this study. Y.K. and W.P. developed the method and analyzed the results. Y.K. wrote the software. J.A. performed the 3C experiments. Z.Z. and J.Z. contributed to methodology design. Y.K., A.T., and W.P. wrote the manuscript. All authors discussed the results and commented on the manuscript.

## Additional information

**Competing interests:** The authors declare no competing interests.

