## [Peer Review File · Nature Communications]

Reviewer #1 (Remarks to the Author):

In this study you analyzed CTCF-mediated loops and furthermore developed a novel tool "Lollipop" to predict CTCF-mediated interactions using genomic and epigenomic features. Based on the required sequencing depth needed to predict such loops the tools could serve as cost-effective alternative to many researchers. Even though the overall approach appears conclusive I have some more questions on the analysis and the prediction of de novo CTCF-mediated loops:

Disease Ontology of GREAT is used to confirm that the super-enhancers within loops are linked to the disease of origin. To my knowledge GREAT annotates (super-)enhancers to genes and based on that disease ontology analysis is carried out.

1) Do you check for each super-enhancer if its annotated gene is in the same loop? Because theoretically it could be annotated to a gene, which is already in a different regulatory domain.

Later in this paragraph you analyze if cell-type specific super-enhancers are within cell-type specific loops.

- 1) Which requirements need to be fulfilled that a super-enhancer is within a loop?
- 2) Does it mean that 100% of the super-enhancer is within one loop?
- 3) Did it occur that you had super-enhancers which go over a boundary of two loops? If yes, how did you handle that?

In the methods part "super-enhancer analysis" you write that you overlap super-enhancers.

1) Please explain more detailed what you considered as overlap (a certain percentage or a couple of base-pairs,...)?

The overall goal of lollipop is to de novo predict CTCF-mediated loops. Suppl. Figure 5 shows that this is very challenging especially by looking at samples with low reads like the K562.

- 1) Please characterize the loops, which are in the category of "no support". Which features do they share in terms of motif orientation, CTCF binding strength, and so on.
- 2) The same would be interesting for loops of the two categories "significant" and "with evidence"

3) Is there a trend that loops from the same category share certain characteristics? And on top of that, if you use for GM12878 all reads, which kind of CTCF-mediated loops could be confirmed.

In the final part of your results you have a look at hubs within each cell type.

1) If you use this information and combine it with the de novo predicted CTCF-mediated loops. Are anchors with a higher number of connections predicted more reliable than other anchors? So what kind of anchors do the loops of "significant", "with evidence" and "no support" have.

Please explain some details regarding software availability.

1) Does the output file contain also characteristics to each loop? This might especially interesting if the, above mentioned, loop categories share certain characteristics.

Small comments

1) In your introduction (line 33) you state that CTCF delimits the boundaries of TADs. However, it is shown that they are enriched at boundaries, but boundaries do not necessarily have CTCF binding.

2) In line 77 are some brackets missing.

3) You write in line 87 that shared loops exhibit a significantly higher interaction strength than cell-type-specific loops. (Suppl. Figure 1a) Please add the p-values to the figure to validate that it is significant.

4) Line 94: citation for the super-enhancer definition is missing

5) I recommend to read the methods part again as there are some typos.

Reviewer #2 (Remarks to the Author):

Kai and colleagues carried out a computational study to predict the genome-wide landscape of CTCF-mediated loops from DNA sequence and ChIPseq data. They showed that a majority of the CTCF-mediated loops are cell-type specific, and these changes may play a role in gene regulation. They further developed a computational method to predict the CTCF loop profiles using a combination of genetic and epigenetic signatures, and obtained very high accuracy. While some of factors are well-known, including CTCF motif and binding strength, other factors are less obvious, including transcription levels and histone modification patterns. This suggests that the genome-wide landscape of CTCF looping can be well-predicted without doing Hi-C or ChIA-PET experiments which remain technically challenging and costly.

In general the paper is clearly written and well-organized, although it is somewhat disappointing that the algorithm is primarily evaluated on the entire set of CTCF loops rather than focusing on cell-type specific changes, which is most interesting. Most of the predictive factors are well known, such as CTCF motif, binding strength, and loop length. Whereas a few other factors are potentially interesting, it is unclear whether they play a significant role in terms of prediction.

Specific comments:

1. Line 75. The authors identified many more CTCF loops in GM12878 than other cell lines, and offered an explanation that this is due to sequencing depth variation. It would be nice to test this hypothesis by down-sampling the GM12878 data.
2. Line 91. The evidence for a regulatory role for cell-type specific CTCF loops is weak. Compared to the motif analysis done in this paper, it would be more direct to test whether the gain or loss of CTCF loops is correlated with gene expression level changes.
3. Line 134. Why is the number of negative controls chosen to be 5 times as many as the positive loops? Is it because a large number of negative samples is needed to estimate the background distribution?
4. Line 143. CTCF motif and binding site information is redundant. It seems plausible that motif information is needed only when ChIPseq data is unavailable. It would be interesting to test the model using just ChIPseq or motif information. It would also be interesting to compare the performance of using ChIPseq binding strength alone vs the full model, in order to assess the utility of the additional information incorporated in the model.
5. Line 149. It is unclear what information from the flanking region is incorporated in the prediction model.
6. Line 182. It is unsurprising that CTCF binding strength is selected as the most important prediction feature. But if this is all it is, the problem becomes trivial. It would be very interesting to know how much of the change of CTCF looping can be attributed to gain or loss of CTCF binding sites. How many loops involve rewiring of existing CTCF sites?
7. Line 223. How well can the model predict cell-type specific loops? These loops are more interesting than the common loops. Is it possible to train a model from one cell-type and apply the model to predict looping in another cell type?

8. Line 230. How robust are the model predictions with respect to the training data? If one uses two different cell lines as training data, and apply each model to predict the looping organization in the third cell line. How much agreement should we expect? This is an important question for practical applications.

9. Line 237-241. For comparison, what is the expected level of overlap if one randomly picks a pair of CTCF sites and call them interacting? If one selects the most stringent predictions, does he/she see a higher degree of overlap?

10. Line 287-289. It is interesting that the CTCF hubs are enriched for cell-type specific functions. What would be the mechanistic basis of such a distinct role of these hubs?

Response to Reviewers' comments:

Thank you for your constructive comments. In the revised manuscript (MS) we have addressed all the issues raised by the reviewers and revised text and figures accordingly. Please find below our point-by-point responses in blue. For your convenience, new figures added in the revised MS are also included in this letter. Relevant changes are highlighted in red in the revised MS.

Reviewer #1:

In this study, you analyzed CTCF-mediated loops and furthermore developed a novel tool "Lollipop" to predict CTCF-mediated interactions using genomic and epigenomic features. Based on the required sequencing depth needed to predict such loops the tools could serve as cost-effective alternative to many researchers. Even though the overall approach appears conclusive I have some more questions on the analysis and the prediction of de novo CTCF-mediated loops:

Major comments

Disease Ontology of GREAT is used to confirm that the super-enhancers within loops are linked to the disease of origin. To my knowledge GREAT annotates (super-)enhancers to genes and based on that disease ontology analysis is carried out.

1) Do you check for each super-enhancer if its annotated gene is in the same loop? Because theoretically it could be annotated to a gene, which is already in a different regulatory domain.

We employed GREAT to perform Disease Ontology analysis (Supplementary Fig. 1e in MS) to all super enhancers (SEs) identified in each cell-type, not just those within loops. The objective of the Disease Ontology analysis was to confirm the role of SEs in cell identity in our context. We now realize that our original text was unclear. Thus, we revised the text to clarify:

"...we ask whether cell-type-specific Super-Enhancers (SEs) are associated with cell-type-specific loops. SEs regulate cell identity, development, and cancer, and CTCF was shown to play a critical role in their hierarchical organization. Motivated by these findings, we first carried out Disease Ontology analysis on SEs for each cell-type using GREAT, confirming that they are linked with the corresponding disease origin..."

The majority of SEs are within loops (99.5% in GM12878, 95% in HeLa, 67% in K562). Even if we only use SEs fully covered by loops, the result of Disease Ontology analysis remains essentially the same (**Figure 1, for reviewer only**), compared to the original result (**Supplementary Fig. 1e in MS**). The reviewer is correct that GREAT links (super-)enhancers to genes and then performs ontology analysis using the associated genes. For SEs within loops, we found that the majority of the SE-gene pairs (95% in GM12878, 95% in HeLa, 84% in K562) are within at least one common loop, suggesting that SEs and the associated genes are in the same regulatory domains.

Figure 1. Disease Ontology analysis of SEs within loops.

Later in this paragraph you analyze if cell-type specific super-enhancers are within cell-type specific loops.

- 1) Which requirements need to be fulfilled that a super-enhancer is within a loop?
- 2) Does it mean that 100% of the super-enhancer is within one loop?

In the analysis of cell-type specific SEs vs cell-type specific loops (**Fig. 1d in MS**), we considered a loop to be associated with a SE as long as they overlap. Therefore, partial coverage was allowed. We revised the main text and the method section to clarify. The revised text, where “enriched within” was replaced with “preferentially associated with”, reads:

“HeLa-specific SEs are preferentially associated with HeLa-specific loops, compared to common SEs (**Fig. 1d** left panel). Similarly, K562-specific SEs are preferentially associated with K562-specific loops compared to common SEs (**Fig. 1d** left panel). The same conclusion was reached when we compared GM12878 vs HeLa as well as GM12878 vs K562 (**Fig. 1d** central and right panels). Taken together, cell-type-specific SEs are more likely to be associated with loops specific to that cell-type....”

To ensure the robustness of our result, we tried the alternative approach of association, counting only loops that completely cover SEs. The conclusion remains unchanged (**Figure 2, for reviewer only**), i.e., cell-type specific SEs are enriched with cell-type specific loops.

Figure 2. Cell-type-specific SEs are enriched with cell-type-specific loops. Only loops fully covering SEs are counted.

3) Did it occur that you had super-enhancers which go over a boundary of two loops? If yes, how did you handle that?

Indeed, **Figure 3 (for reviewer only)** shows such an example, where SE1 and SE3 (highlighted in red) go over the boundaries of different loops. In the case of a SE going over a boundary of two loops, we associate both loops with the SE. In general, we adopted the strategy of associating a loop with a SE if the loop region overlaps with the SE. We think that this strategy would not introduce any bias for or against any subset of SEs (i.e., cell-type specific SEs or shared SEs). The robustness of our conclusion is demonstrated by results shown in **Figure 2**, where we only counted loops fully covering SEs.

Figure 3. An example showing SEs (highlighted) that go over boundaries of loops.

In the methods part "super-enhancer analysis" you write that you overlap super-enhancers.

1) Please explain more detailed what you considered as overlap (a certain percentage or a couple of base-pairs)?

If the genomic regions of two SEs overlap by at least 1 bp, these SEs are considered overlapped. We now clarify this point in the method section.

The overall goal of lollipop is to de novo predict CTCF-mediated loops. Suppl. Figure 5 shows that this is very challenging especially by looking at samples with low reads like the K562.

- 1) Please characterize the loops, which are in the category of "no support". Which features do they share in terms of motif orientation, CTCF binding strength, and so on.
- 2) The same would be interesting for loops of the two categories "significant" and "with evidence"

We characterized three categories of *de novo* predicted loops in terms of motif orientation pattern, CTCF binding strength, loop length and other features. Loops in the 'Significant' and 'With evidence' categories are more likely to have convergent motifs (**Figure 4a below, New Supplementary Fig. 5b in MS**). However, distributions of CTCF binding, loop length (**Figure 4b, for reviewer only**) and other features (data not shown) exhibit no substantial differences. CTCF binding in the "significant" category exhibits slightly higher binding intensity. These results are incorporated in the revised MS.

Figure 4. Characterization of the “Significant”, “With evidence” and “No support” loops. **(a)** Distribution of motif orientation. Loops in the ‘With evidence’ and ‘Significant’ categories are enriched with convergent motif orientation compared with loops in the ‘No support’ category (P -value < $1e-100$ for all pair-wise comparison, Chi-squared test). **(b)** Distribution of CTCF binding intensity (“avg_CTCF”, average of binding intensity of CTCF at the two anchors) and loop length.

3) Is there a trend that loops from the same category share certain characteristics? And on top of that, if you use for GM12878 all reads, which kind of CTCF-mediated loops could be confirmed.

As elaborated in the previous comment, we found that the predicted loops that find support in ChIA-PET data are more likely to be convergent than loops in the ‘No support’ category. We observed no apparent differences in other characteristics.

Regarding the confirmation of *de novo* predictions, we would like to point out that the predicted loops in the “significant” and “with evidence” categories are supported by ChIA-PET data by definition. As shown in the original analyses, the loops in the “No support” category are supported by Hi-C data (**Fig. 6a in MS**).

To further confirm these predictions, we examined the contact frequency from Hi-C data of the predicted loops in all three categories in GM12878 (all reads). As shown in **Figure 5 (New**

Supplementary Fig. 5e in MS), all exhibit significantly higher contact frequency than those of random genomic regions. Furthermore, the stronger the support from ChIA-PET, the higher the contact frequency from Hi-C. (P -value $<1e-300$ for both “With evidence” vs. “No support” and “Significant” vs. “With evidence”, Mann Whitney U test). This result is incorporated in the revised MS.

Figure 5. Hi-C validation of “No support”, “With evidence” and “Significant” loops. Blue: predicted loops. Green, random genomic regions. For each category, the set of random regions were chosen with matching size and length distribution.

In the final part of your results you have a look at hubs within each cell type.

1) If you use this information and combine it with the de novo predicted CTCF-mediated loops. Are anchors with a higher number of connections predicted more reliable than other anchors? So what kind of anchors do the loops of "significant", "with evidence" and "no support" have.

Indeed, we found the predicted hubs have a higher predictive accuracy than non-hubs (**Figure 6a, for reviewer only**). Regarding the second part of the question, we calculated the percentage of hubs in the three categories of loops: ‘Significant’, ‘With evidence’, ‘No support’, and found no clear trend (**Figure 6b, for reviewer only**). This is understandable because the loops emanated from a hub can belong to different categories of loops.

Figure 6. (a) Comparison of the anchor prediction accuracy for hubs (top 10% of predicted anchors ranked by degree of connection) and non-hubs (bottom 10% of predicted anchors). (b) The percentage of hubs in “Significant”, “With evidence”, and “No support” loops.

Please explain some details regarding software availability.

1) Does the output file contain also characteristics to each loop? This might especially interesting if the, above mentioned, loop categories share certain characteristics.

We agree that the characteristics to each loop are potentially useful for further analysis. Our default output file contains only the anchor locations and looping probability for each loop. We have added an option to the script that enables the generation of loops with associated characteristics (i.e., anchor locations, looping probabilities, and values for all features).

Small comments

1) In your introduction (line 33) you state that CTCF delimits the boundaries of TADs. However, it is shown that they are enriched at boundaries, but boundaries do not necessarily have CTCF binding.

We have changed the text to be "...CTCF is bound at loop anchors and enriched at the boundaries of Topologically Associating Domains (TADs)...".

2) In line 77 are some brackets missing.

We fixed the brackets.

3) You write in line 87 that shared loops exhibit a significantly higher interaction strength than cell-type-specific loops. (Suppl. Figure 1a) Please add the p-values to the figure to validate that it is significant.

P-values were added to the figure (**Supplementary Fig. 1c in MS**).

4) Line 94: citation for the super-enhancer definition is missing

We added the citation for super-enhancer definition (Hnisz, D et.al 2013¹).

5) I recommend to read the methods part again as there are some typos.

We proofread again and corrected typos in the Methods section.

Reviewer #2:

Kai and colleagues carried out a computational study to predict the genome-wide landscape of CTCF-mediated loops from DNA sequence and ChIPseq data. They showed that a majority of the CTCF mediated loops are cell-type specific, and these changes may play a role in gene regulation. They further developed a computational method to predict the CTCF loop profiles using a combination of genetic and epigenetic signatures, and obtained very high accuracy. While some of factors are well-known, including CTCF motif and binding strength, other factors are less obvious, including transcription levels and histone modification patterns. This suggests that the genome-wide landscape of CTCF looping can be well-predicted without doing Hi-C or ChIA-PET experiments which remain technically challenging and costly.

In general, the paper is clearly written and well-organized, although it is somewhat disappointing that the algorithm is primarily evaluated on the entire set of CTCF loops rather than focusing on cell-type specific changes, which is most interesting. Most of the predictive factors are well known, such as CTCF motif, binding strength, and loop length. Whereas a few other factors are potentially interesting, it is unclear whether they play a significant role in terms of prediction.

We thank the reviewer for the valuable critiques and suggestions. Overall,

1) We evaluated Lollipop's performance on predicting cell-type-specific loops and found that Lollipop performs well. Please see the response to comment #7 for details.

2) To better understand the predictive significance of features, we compared the performance of models using well-known features only (i.e., CTCF motif orientation, CTCF and Cohesin binding at anchors, and loop length) with models using all features. We found that incorporation of additional features improves AU-PR by about 0.1 (0.11 in GM12878, 0.06 in HeLa and 0.12 in K562), demonstrating their utility. Another benefit is that the additional features provide predictive power when CTCF and Cohesin ChIP-seq data are not available, as many features contain redundant information (**Fig. 4d and Supplementary Fig. 3f in MS**). Please see the response to comment #4 for additional details.

Specific comments:

1. Line 75. The authors identified many more CTCF loops in GM12878 than other cell lines, and offered an explanation that this is due to sequencing depth variation. It would be nice to test this hypothesis by down-sampling the GM12878 data.

We performed a systematic down-sampling analysis of the GM12878 ChIA-PET data, confirming that a higher sequencing depth enables detection of more significant loops (**Figure 7 below, New Supplementary Fig. 1a in MS**). However, there may be other factors contributing to loop identification, including (1) experimental methods, as K562 ChIA-PET dataset was generated by a short-read protocol², while GM12878/HeLa datasets³ were produced by a long-read protocol, and (2) cell-type, as different cell-types may have different “3C-seq-ability” due to different digestion efficiency and other factors⁴. To make our statement more precise, we have changed the text to “Of note, GM12878 library has higher sequencing depth, which may contribute to the higher number of identified loops and cell-type-specific loops...”

Figure 7. Down-sampling analysis of GM12878 ChIA-PET data. The star denotes the point for 15% down-sampling.

2. Line 91. The evidence for a regulatory role for cell-type specific CTCF loops is weak. Compared to the motif analysis done in this paper, it would be more direct to test whether the gain or loss of CTCF loops is correlated with gene expression level changes.

Following the reviewer’s suggestion, we directly tested the expression changes of genes associated with cell-type-specific loops. We found that genes associated with cell-type-specific loops have higher expression levels in the respective cell-type. In contrast, there is no significant difference in expression for genes associated with shared loops (**Figure 8 below, New Supplementary Fig. 1f**). This result is incorporated in the MS.

We would like to note that we carried out several analyses to support the notion that cell-type-specific loops play a regulatory role. In addition to the motif orientation analysis, we showed that cell-type-specific SEs are more likely to be associated with loops specific to that cell-type (**Fig. 1d in MS**). We also showed that differentially expressed genes (DEGs) are more frequently associated with cell-type-specific loops (**Supplementary Fig. 1g in MS**), and those associated DEGs are linked to distinct pathways (**Fig. 1e and Supplementary Fig. 1h-i in MS**)

Figure 8. Boxplots showing expression levels of genes associated with cell-type-specific and shared loops in pair-wise comparison of cell-types. A gene is associated with a loop if its promoter region (TSS +/- 2kb) is inside the loop. P -values were calculated using Mann-Whitney U test.

3. Line 134. Why is the number of negative controls chosen to be 5 times as many as the positive loops? Is it because a large number of negative samples is needed to estimate the background distribution?

Yes. A similar approach was adopted by Whalen S et.al 2016⁵ in predicting promoter-enhancer loops.

4. Line 143. CTCF motif and binding site information is redundant. It seems plausible that motif information is needed only when ChIPseq data is unavailable. It would be interesting to test the model using just ChIPseq or motif information. It would also be interesting to compare the performance of using ChIPseq binding strength alone vs the full model, in order to assess the utility of the additional information incorporated in the model.

To examine whether the CTCF motif and binding site information are redundant in predictive power, we followed the reviewer's suggestion. The results in GM12878 are summarized in **Figure 9 (New Supplementary Fig. 3g in MS)**. The performance of CTCF motif or CTCF binding alone are AU-PR=0.5 and 0.45 respectively, while CTCF motif+ChIP-seq model substantially improves the performance (AU-PR=0.61). Moreover, the all-feature model (AU-PR=0.86) performs significantly better than the CTCF motif+ChIP-seq model and the model with only ChIP-seq binding. While CTCF motif and binding site share information, we believe that they are not completely redundant because (1) Not all CTCF binding sites have motifs. We found that ~35% of CTCF binding sites lack significant motifs (34% for GM12878, 37% for HeLa, 33% for K562), and (2) Information on motif orientation cannot be derived from CTCF binding.

Of note, **Fig. 9** includes a result demonstrating the benefit of features other than well-known ones, as they can compensate in predictive power when not all well-known features are available. As shown, the model without features derived from CTCF and Cohesin ChIP-Seq performs reasonably well.

Figure 9. PR curve for comparison of performance of models. CTCF motif-only features include motif orientation pattern, motif strength and conservation score on anchors. ChIP-seq features include CTCF binding strength at the two anchors. Features without CTCF and RAD21 denotes all features except those derived from CTCF and RAD21 ChIP-seq.

5. Line 149. It is unclear what information from the flanking region is incorporated in the prediction model.

The information from the flanking regions include signal intensity of histone modifications, DNase, CTCF and Cohesin (all in RPKM). The consideration of the flanking features is motivated by the insulating role of CTCF, in which CTCF binding could prevent the spreading of heterochromatin from one side to the other. Therefore, we reasoned that the potential imbalance of the epigenetic marks, such as H3K27me3 and H3K9me3, might contribute in predicting the loop pattern. To clarify this information, we revised the manuscript as follows: "In addition, given the insulator role

of CTCF, we reasoned that the potential imbalance of signal intensities on the two sides of CTCF anchors might contribute to the prediction. Therefore, we used the DNase and ChIP-seq signals on the flanking regions as features.”

6. Line 182. It is unsurprising that CTCF binding strength is selected as the most important prediction feature. But if this is all it is, the problem becomes trivial. It would be very interesting to know how much of the change of CTCF looping can be attributed to gain or loss of CTCF binding sites. How many loops involve rewiring of existing CTCF sites?

We agree that this is a very interesting question. It was addressed in the original MS (**Fig. 1c in MS**). In short, rewiring is substantial, accounting for 24% to 44% of the cell-type-specific loops.

7. Line 223. How well can the model predict cell-type specific loops? These loops are more interesting than the common loops. Is it possible to train a model from one cell-type and apply the model to predict looping in another cell-type?

With regard to the first question (i.e., “How well can the model predict cell-type specific loops?”), we evaluated the performance using the loops specific to the target cell-type. The performance is only slightly lower than that of the full set of loops (**Figure 10 below, New Fig. 5b and Supplementary Fig. 4b in MS**). For example, in HeLa to GM12878, AU-PR is 0.78 (cell-type-specific) vs 0.81 (full). This result suggests that Lollipop predicts cell-type-specific loops well.

Figure 10. Performance evaluation of cell-type-specific loops in terms of (left) ROC and (right) PR curves. In the title of each subplot, ‘A to B’ denotes training the model in A and evaluating the model in B, where the positive loops were chosen as those specific to B and negative loops were random pairings of CTCF binding sites and were 5 times as abundant as the positive loops.

The answer to the second question (i.e., “Is it possible to train a model from one cell-type and apply the model to predict looping in another cell type”) is yes, as demonstrated by our original analysis (**Fig. 5a and Supplementary Fig. 4a in MS**) and analysis above (**Figure 10**). Moreover, we carried out an evaluation of *de novo* predictions across cell-types. Specifically, we trained the model in HeLa and K562 cells, and applied them to make genome-wide predictions in GM12878. We then used the ChIA-PET loops observed in GM12878 to evaluate the performance, as the GM12878 ChIA-PET library has the highest sequencing depth and hence allow for better evaluations. We found that the performance of models trained from HeLa and K562 are on-par with and slightly lower than the performance of model trained from the down-sampled GM12878 data (**Figure 11, for reviewer only**).

Figure 11. Performance comparison of *de novo* genome-wide predictions in GM12878 using different models.

8. Line 230. How robust are the model predictions with respect to the training data? If one uses two different cell lines as training data, and apply each model to predict the looping organization in the third cell line. How much agreement should we expect? This is an important question for practical applications.

To address this question, we compared the *de novo* predictions in GM12878 using the models trained from HeLa and K562 cells. For all pairs of CTCF binding sites in GM12878, the looping probabilities predicted from HeLa and K562 models are concordant (**Figure 12a below, New Supplementary Fig. 5h**), with a Pearson correlation of 0.88 (P -value $<1e-300$). If we use the probability cut-off value of 0.5 to call loops, the predicted loops from HeLa and K562 models significantly overlap (**Figure 12b below, New Supplementary Fig. 5i**). Furthermore, the loops predicted by both models are much more likely to find support in the GM12878 ChIA-PET data, compared with loops predicted only in HeLa or K562 models alone (**Figure 12c, for reviewer only**). Additionally, results of features importance analysis in different cell-types (**Fig. 4a and Supplementary Fig. 3a in MS**) are also similar, suggesting the robustness of our approach.

a

b

c

Figure 12. (a) Scatter plot with hexagonal binning showing the predicted looping probability values of all pairs of CTCF binding in GM12878 from HeLa (x-axis) and K562 (y-axis) models. (b) The overlap of loops in GM12878 predicted from HeLa and K562 models. (c) Overlap of loops in GM12878 predicted from HeLa and K562 models with ChIA-PET result of GM12878.

9. Line 237-241. For comparison, what is the expected level of overlap if one randomly picks a pair of CTCF sites and call them interacting? If one selects the most stringent predictions, does he/she see a higher degree of overlap?

The results are summarized in **Figure 13 (New Supplementary Fig. 5a in MS)**. The first column addresses the first question on the expected overlap in background. “Significant” loops (i.e., loops called from ChIA-PET data) accounts for less than 6% of all pairings of CTCF sites. (6%, 2%, 1% for GM12878, HeLa and K562 respectively). The other columns address the second part of the question. Indeed, as the loop-calling stringency increases, higher percentage of predicted loops find support in ChIA-PET data (blue + orange), and higher percentage falls into the “significant” category (blue). This result is incorporated in the MS.

Figure 13. Overlap of *de novo* predicted loops with ChIA-PET results at varying prediction stringencies. The prediction stringency (probability cut-off value) and the number of predicted loops are denoted at the bottom of each column.

10. Line 287-289. It is interesting that the CTCF hubs are enriched for cell-type specific functions. What would be the mechanistic basis of such a distinct role of these hubs?

One possibility is that the CTCF hubs are associated with distinct sets of transcription factors in different cell-types. To test this hypothesis, we examined the enrichment of transcription factor motifs at the CTCF hubs. The enrichment pattern is cell-type-specific (**Figure 14 below, New Supplementary Fig. 6b in MS**). Notably, ZFX, ARNT, and MZF1 are only enriched in GM12878 and K562 cells, consistent with their hematopoietic origin^{6, 7, 8}. This result is incorporated in the MS.

Figure 14. Heatmap showing the enrichment pattern of motifs of transcription factors at the CTCF hubs from the three cell-types. Motif enrichment analysis was performed using the SeqPos tool⁹.

To further improve the manuscript, the following revisions have been made.

- 1) We carefully proofread the entire MS and made typo corrections and clarifications.
- 2) We added an illustration (**New Supplementary Fig. 1b in MS**) of the two types of cell-type-specific loops due to either cell-type-specific CTCF binding or rewiring.
- 3) We updated the implementation of the gene expression feature in Lollipop and re-ran the analyses. All major results remain unchanged, although there are slight changes in the feature ranking in **Fig.4** and **Supplementary Fig. 3 in MS**.

1. Hnisz D, *et al.* Super-enhancers in the control of cell identity and disease. *Cell* **155**, 934-947 (2013).
2. Consortium EP. An integrated encyclopedia of DNA elements in the human genome. *Nature* **489**, 57-74 (2012).
3. Tang Z, *et al.* CTCF-Mediated Human 3D Genome Architecture Reveals Chromatin Topology for Transcription. *Cell* **163**, 1611-1627 (2015).
4. Stadhouders R, *et al.* Multiplexed chromosome conformation capture sequencing for rapid genome-scale high-resolution detection of long-range chromatin interactions. *Nat Protoc* **8**, 509-524 (2013).
5. Whalen S, Truty RM, Pollard KS. Enhancer-promoter interactions are encoded by complex genomic signatures on looping chromatin. *Nat Genet* **48**, 488-496 (2016).
6. Galan-Caridad JM, *et al.* Zfx controls the self-renewal of embryonic and hematopoietic stem cells. *Cell* **129**, 345-357 (2007).
7. Morris JF, *et al.* The myeloid zinc finger gene, MZF-1, regulates the CD34 promoter in vitro. *Blood* **86**, 3640-3647 (1995).
8. Adelman DM, Maltepe E, Simon MC. Multilineage embryonic hematopoiesis requires hypoxic ARNT activity. *Genes Dev* **13**, 2478-2483 (1999).
9. He HH, *et al.* Nucleosome dynamics define transcriptional enhancers. *Nat Genet* **42**, 343-347 (2010).

Reviewer #1 (Remarks to the Author):

The authors addressed all my comments in details. Thus, I do not have further concerns.

Reviewer #2 (Remarks to the Author):

The authors have done an excellent job in revising their manuscript and satisfactorily addressed all my previous concerns.